# PBQ-Enhanced QUIC: QUIC with Deep Reinforcement Learning Congestion Control Mechanism

**DOI:** 10.3390/e25020294

**Published:** 2023-02-04

**Authors:** Zhifei Zhang, Shuo Li, Yiyang Ge, Ge Xiong, Yu Zhang, Ke Xiong

**Affiliations:** 1Engineering Research Center of Network Management Technology for High Speed Railway of Ministry of Education, School of Computer and Information Technology, Beijing Jiaotong University, Beijing 100044, China; 2Collaborative Innovation Center of Railway Traffic Safety, Beijing Jiaotong University, Beijing 100044, China; 3National Engineering Research Center of Advanced Network Technologies, Beijing Jiaotong University, Beijing 100044, China; 4China Software and Technical Service Co., Ltd., Beijing 100081, China; 5Institute of Economics and Energy Supply and Demand, State Grid Energy Research Institute Co., Ltd., Beijing 102209, China

**Keywords:** QUIC, congestion control, deep reinforcement learning, BBR

## Abstract

Currently, the most widely used protocol for the transportation layer of computer networks for reliable transportation is the Transmission Control Protocol (TCP). However, TCP has some problems such as high handshake delay, head-of-line (HOL) blocking, and so on. To solve these problems, Google proposed the Quick User Datagram Protocol Internet Connection (QUIC) protocol, which supports 0-1 round-trip time (RTT) handshake, a congestion control algorithm configuration in user mode. So far, the QUIC protocol has been integrated with traditional congestion control algorithms, which are not efficient in numerous scenarios. To solve this problem, we propose an efficient congestion control mechanism on the basis of deep reinforcement learning (DRL), i.e., proximal bandwidth-delay quick optimization (PBQ) for QUIC, which combines traditional bottleneck bandwidth and round-trip propagation time (BBR) with proximal policy optimization (PPO). In PBQ, the PPO agent outputs the congestion window (CWnd) and improves itself according to network state, and the BBR specifies the pacing rate of the client. Then, we apply the presented PBQ to QUIC and form a new version of QUIC, i.e., PBQ-enhanced QUIC. The experimental results show that the proposed PBQ-enhanced QUIC achieves much better performance in both throughput and RTT than existing popular versions of QUIC, such as QUIC with Cubic and QUIC with BBR.

## 1. Introduction

At present, computer networks remain the essential platform for information interaction, where the transport layer plays an influential role. The emergence of modern applications, such as video live broadcast and Internet of Things (IoT), has imposed higher demands on throughput, packet loss rate, and network delay. The development of wireless transmission technologies such as 5G and WiFi has made the network environment even more complex. The Transmission Control Protocol (TCP), as a widely used protocol, experiences problems such as large handshake delay, head-of-line (HOL) blocking, and protocol solidification, which increasingly affect network performance. Compared with TCP, the User Datagram Protocol (UDP) is more efficient for real-time transmission but lacks reliability.

In 2012, Google proposed the Quick UDP Internet Connection (QUIC) protocol [1], which realizes orderly, quick, and reliable transport services in user mode based on UDP. The QUIC protocol reduces the handshake latency to zero round-trip time (RTT) by caching ServerConfig. Moreover, the QUIC protocol natively supports multiplexing techniques where streams on the same connection do not influence each other, which solves the HOL blocking problem. Additionally, the QUIC protocol decouples the congestion control algorithm from the protocol stack, which is more flexible than TCP. Currently, the QUIC protocol has become one focus of researchers in both academia and industry.

For both TCP and QUIC protocols, congestion control is one of the core mechanisms. The basic logic of TCP congestion control contains congestion sensing and the corresponding disposal pattern, which improves the network utilization by estimating the bandwidth-delay product (BDP) of the link and adjusting the send rate of clients. Since the first congestion control algorithm, Tahoe [2], was proposed, there have been dozens of congestion control algorithms developed that are suitable for different scenarios. The operation rules of traditional congestion control algorithms are shown in Figure 1. Congestion control algorithms work on the sender, sensing the network state and changing the congestion window or transmission rate. Currently widely used control algorithms are heuristic and cannot be optimally executed in dynamic and changeable network environments.

Recently, researchers have started to design TCP congestion control algorithms using machine learning algorithms. There are numerous versions of TCP based on supervised learning, such as TCP with DeePCCI [3] and TCP with LSTM-PTI [4]. Naturally, supervised-learning-based algorithms are only used for passive congestion identification, and training them requires a great deal of labeled data. As the network environment changes, so do the network characteristics. In this case, the supervised learning congestion control algorithm is not suitable for the changing network environment and may no longer be effective. In light of the advantages of the reinforcement learning (RL) method in sequence decision [5,6], researchers all over the world have been trying to apply it to congestion control. So far, some well-known congestion control methods, such as Remy [7], performance-oriented congestion control (PCC)-Vivace [8], Q-learning TCP (QTCP) [9], Orca [10] and Aurora [11] algorithms have been proposed for the TCP protocol. For the QUIC protocol, researchers worldwide have proposed some modified versions of QUIC, i.e., QUIC-go [12], Microsoft QUIC (MsQUIC) [13], Modified QUIC [14], and Quiche [15]. QUIC-go is an implementation of the QUIC protocol in Go. It keeps up to date with the latest request for comments (RFC) and is easy to deploy. MsQUIC is a Microsoft implementation of the QUIC protocol. It optimizes the maximal throughput and minimal latency. Modified QUIC proposes a modification to the handshaking mechanism to minimize the time required to update the CWnd, which results in a smooth variation in the CWnd. This makes modified QUIC protocol easy to deploy and achieves better performance in terms of throughput. Quiche is an implementation of the QUIC protocol in Rust and C. It performs well across different applications, such nginx and curl.

However, for these popular versions of QUIC, only heuristic congestion control algorithms are applied. Most heuristic congestion control algorithms commonly perceive the network state based on simple network models and adopt a fixed strategy, i.e., shrinking the window when packet loss occurs or RTT increases, and augmenting the window when an acknowledge character (ACK) is received. Such a simple strategy makes heuristic congestion control algorithms unable to achieve optimal results in dynamic and changeable network environments. For example, in the WiFi scenario, where obstacles and human activity affect network quality, packet loss events do not imply the occurrence of congestion. However, heuristic congestion control algorithms cannot distinguish that. This issue leads to poor performance of the QUIC protocol in terms of throughput and latency when the network settings change. Therefore, to enable the QUIC to achieve better performance in different network environments, we tried to design DRL-enhanced QUIC using the merits of DRL in environment awareness and decision making.

The contributions of this work can be summarized as follows:First, we developed a novel congestion control mechanism, referred to as proximal bandwidth-delay quick optimization (PBQ) by combining proximal policy optimization (PPO) [16] with traditional BBR [17]. It is able to effectively improve the convergence speed and link stability during the training phase. We then applied the presented PBQ to the QUIC protocol and formed a new version of QUIC, i.e., PBQ-enhanced QUIC, which aims to enhance its adaptivity and throughput performance.Second, for the purpose of reducing the action space and establishing the connection of the values of the congestion window (CWnd) for each interaction, we used continuous action ratio as the output action of PBQ’s agent. Additionally, in the design of the utility function, we used a relatively simple formulation of the objective function as the optimization objective and introduced a delay constraint. By doing so, our proposed PBQ-enhanced QUIC achieves higher throughput and maintains a low RTT.Third, we built a reinforcement learning environment for QUIC on the network simulation software ns-3, where we trained and tested PBQ-enhanced QUIC. The experimental results showed that our presented PBQ-enhanced QUIC achieves much better RTT and throughput performance than existing popular versions of QUIC, such as QUIC with BBR and QUIC with Cubic [18].

The organization of the rest of this paper is as follows: Section 2 describes the QUIC protocol, including the handshake phase, multiplexing, connection migration, and congestion control. Section 3 gives the design of PBQ and PBQ-enhanced QUIC. Section 4 introduces the training link and testing link and evaluates the performance of PBQ-enhanced QUIC compared with that of QUIC using different congestion control algorithms. Finally, Section 5 gives the conclusions.

## 2. QUIC Protocol

### 2.1. The Basic Concept of QUIC

The QUIC protocol is a secure and reliable data transfer protocol first proposed by Google in 2012, which was released as a standardized version of QUIC, RFC 9000 [19], by the Internet Engineering Task Force (IETF). The bottom layer of QUIC is UDP, which makes it compatible with current network protocols. At the same time, the QUIC protocol is compatible with traditional congestion control algorithms in TCP. The QUIC protocol has numerous improvements over TCP, mainly in handshaking, multiplexing, and connection migration.

### 2.2. Handshake

TCP is a plaintext transport protocol and transport layer security (TLS) is required to implement data encryption transmission. In TLS 1.2, two RTTs are required for the TLS handshake phase and three RTTs for the total handshake delay. Under TLS 1.3, the TLS handshake phase requires one RTT, and the handshake phase still requires two RTTs.

Comparatively, the QUIC protocol optimizes the handshake process. In the handshake phase, QUIC incorporates the transmission parameters into the encryption part. When the first connection is established, the client sends authentication- and encryption-related information to the server. It takes only one RTT to establish a connection. Alternatively, we can use quantum key distribution to replace the original encryption information exchange [20]. When the connection is established once again, the client uses the preshared key to establish an encrypted connection with the server during zero RTT.

### 2.3. Multiplexing

Multiplexing is multiple streams on a connection at the same time, commonly on a Hyper Text Transfer Protocol (HTTP) stream. For HTTP/2, a TCP connection can support multiple HTTP streams. However, HOL blocking in the TCP protocol causes interference between HTTP streams, which affects their performance. In advance, multiple streams can be created on a single QUIC connection, which reduces the handshake frequency. The QUIC protocol is implemented based on UDP. Streams on the same QUIC connection are independent of each other, which solves the HOL blocking problem.

### 2.4. Connection Migration

One TCP connection is known to be identified by a quintuple, i.e., <$IP_{source}$, $PORT_{source}$, $IP_{dest}$, $PORT_{dest}$>, where $IP_{source}$ denotes the source IP address, $PORT_{source}$ denotes the source port, $IP_{dest}$ denotes the destination IP address, and $PORT_{dest}$ denotes the destination port [21]. If one term in the quadruple changes, the connection is broken. By using a 64-bit random ID as the connection identifier, the QUIC protocol avoids the effect of network switching. The simple process of connection migration is shown in Figure 2.

Particularly, before the client’s IP changes, the endpoints communicate via a nondetection packet. After IP changes, the path detection is performed to verify the reachability before connection migration starts. If the path detection fails, the connection migration cannot be performed; otherwise, a fresh connection is established. The IP address is verified between the client and the server, and the endpoint holding the latest IP address of the peer migrates. When migration occurs, the congestion control part and the reliable transport protocol estimation part of the path need to be reset. After the connection migrates, it sends a nonprobe packet.

### 2.5. Congestion Control in QUIC

The QUIC protocol is a reliable UDP-based data transfer protocol, which makes it compatible with existing network protocols. The congestion control algorithm in TCP is implemented in kernel mode, and if an upgrade is performed, the kernel needs to be recompiled. However, the QUIC protocol straddles kernel mode and user mode, and its congestion control part is in user mode. Thus, it can be easily upgraded. In particular, the QUIC protocol supports the configuration of different congestion control algorithms for applications, making it possible to optimize for specific applications.

## 3. The PBQ-QUIC

In this section, we describe PBQ and its application to QUIC to form the PBQ-enhanced QUIC.

### 3.1. The Basic Idea of PBQ

For clarity, Figure 3 shows the workflow of PBQ. PBQ combines PPO with BBR to accurately identify the network state. It can effectively identify congestion and packet loss events. PBQ is divided into three main parts: Monitoring, Decision, and Pacer modules. The Monitoring module collects environment state and sends it to the Decision module. The Decision module outputs actions, including $a_{t}$ and the pacing rate, according to the network state. The Pacer module distributes actions to the corresponding senders. In the Decision module, the Controller distributes the network state to the PPO and the BBR. The PPO part outputs $a_{t}$ according to $s_{t}$ using the new policy and sends it to the environment through Pacer. The replay memory stores the experience of past interactions. After multiple interactions, the policy networks and the value network are updated using the past interaction experience. Inspired by Orca, our Decision module adopts a two-level regulation mechanism, which is shown in Figure 4. The underlying BBR algorithm performs a classical decision-making behavior driven by ACK. The DRL agent evaluates the network congestion and predicts the BDP according to the state output by the Monitoring module.

Monitoring Module

We set the network state collection interval to 100 ms, and the designed state quantity is shown in Table 1. Because the interval is not strictly equal to 100 ms, we also count the intervals. The remaining states are the statistics within each interval.

Reward Function Design

We trained PBQ using a linear function with constraints. First, we define a linear utility function as $\alpha *deliveryRate_{t}-\beta *packetLoss_{t}$, where α denotes the coefficient of $deliveryRate_{t}$, $deliveryRate_{t}$ denotes the delivery rate in time *t*, β denotes the coefficient of $packetLoss_{t}$, and $packetLoss_{t}$ denotes the packet loss rate in time *t*. Then, we formulate an optimization problem to maximize the linear utility function under a given RTT constraint, i.e., $RTT_{t}\leq minRTT_{t}\;,t=1,2,3\ldots ,n$, where $RTT_{t}$ denotes the last RTT in time *t*, and $minRTT_{t}$ denotes the minimum RTT from the establishment of the connection to time *t*.
(1)$$\begin{array}{cc} \; & \max\;\alpha *deliveryRate_{t}-\beta *packetLoss_{t}\\ \; & \begin{array}{ccc} s.t. & RTT_{t}\leq minRTT_{t}\;, & t=1,2,3\ldots ,n \end{array}. \end{array}$$ We use the maximization objective as the base reward function:(2)$$R_{t}=\alpha *deliveryRate_{t}-\beta *packetLoss_{t}$$ With the delay constraint, the final reward function is
(3)$$\begin{array}{c} R_{t}=\alpha *deliveryRate_{t}-\beta *packetLoss_{t}\\ R_{t}=\left\{\begin{array}{ccc} R_{t} & \; & RTT_{t}\leq \gamma *minRTT_{t}\\ R_{t}-\eta \; & \; & RTT_{t}>\gamma *minRTT_{t} \end{array}\right. \end{array}$$
where γ denotes the penalty threshold, and η denotes the penalty term when $RTT_{t}>\gamma *minRTT_{t}$.

Action Design

Reinforcement learning action types can be divided into two categories, discrete and continuous actions. We first used discrete actions, but analyzing the experimental results, we found that when using discrete actions, the output of the agent considerably fluctuated, and it was difficult to achieve better performance in the initial stages of the interaction. In general, the discrete action does not favor the early stability of the network links. The final solution in this study draws on the additive increase multiplicative decrease (AIMD) [22] idea in traditional congestion control, and we set the action output as CWndRatio. The mapping relationship between CWndRatio and the congestion window is as follows:(4)$$newCWnd=CWnd*2^{CWndRatio}$$ The relationship between the values of CWnd for each interaction was established, which reduces the fluctuation of the action and ensures the excellent performance of the model.

Learning Algorithm for PBQ

For the DRL agent, PPO is employed, which is a classical actor-critic algorithm. We used tanh as the activation function in both actor and critic neural networks. Because the state transitions in traditional congestion algorithms are not complex, we preferred to build a simpler neural network and train on it. Finally, we constructed a three-layer neural network, where the hidden layer contains 64 neurons.

For clarity, the PBQ training phase is summarized in Algorithm 1. On lines 1–2 of Algorithm 1, the training episode Episodes is set and the replay memory *D* is initialized, which is used to store state, action and reward. The policy parameters $\theta_{new}$ and value parameters ϕ are also initialized with random weights. We set policy parameters $\theta_{old}$ equal to $\theta_{new}$. Then, on lines 4–16, at each episode, we first reset the environment and obtain the initial state $s_{0}$ and initial reward $r_{0}$. We use PPO to regularly output $\alpha_{t}$ according to the new policy $\pi_{\theta_{new}}$. We map $\alpha_{t}$ to $CWnd_{t}$, which is also the estimated link BDP. Then, $CWnd_{t}$ is preformed into environment. In each step, PBQ collects network states $s_{t}$ and $s_{t+1}$, action value $a_{t}$, and the corresponding reward $r_{t}$, and updates the replay memory *D*. The policy parameters $\theta_{new}$ and value functions ϕ are updated when the number of steps accumulates to the policy update threshold update_timestep. The underlying BBR method is driven by ACK to control the specific pacing rate of the clients.
**Algorithm 1** PBQ’s learning algorithm.1:Initialize replay memory *D* and training episode Episodes2:Initialize policy parameters $\theta_{new}$, value parameters ϕ, $\theta_{old}\leftarrow \theta_{new}$3:**for**episode = 1 to Episodes **do**4:     Reset environment, obtain initial state $s_{0}$ and initial reward $r_{0}$5:     i←16:     **for** t = 1 to steps **do**7:         **if** time to play PPO action **then**8:             Run policy $\pi_{\theta_{new}}$, obtain PPO action $\alpha_{t}$9:             Map $\alpha_{t}$ to $CWnd_{t}$, $CWnd_{t}=CWnd_{t-1}*2^{\alpha_{t}}$10:           Perform $CWnd_{t}$11:        **else**12:            Play a BBR action, perform $pacingRate_{t}$13:        **end if**14:        Collect $\left\{s_{t},a_{t},r_{t},s_{t+1}\right\}$, update *D*15:        t=t+116:        **if** t%update_timestep==0 **then**17:            $\theta_{old}\leftarrow \theta_{new}$, update $\theta_{new}$ and ϕ18:        **end if**19:      **end for**20:**end for**

Pacer

The Pacer module is responsible for distributing the actions output by the Decision module to the corresponding clients. The action distribution is driven by an update callback for the CWnd and an update callback for the pacing rate. For pacing rate, when the client receives an ACK or packet loss occurs, the underlying BBR adjusts the pacing rate and triggers a pacing rate update callback. For CWnd, the Decision module receives the updated network state from the Monitoring module; the RL part outputs α and calculates the current CWnd; then, the callback function is called to update the CWnd on the client via Pacer module. As shown in Figure 5, the Pacer module distributes actions to the corresponding clients based on their ids.

### 3.2. PBQ-Enhanced QUIC

Figure 6 shows the differences in congestion control between PBQ-enhanced QUIC and traditional QUIC. QUIC uses Cubic as the default congestion control algorithm. Cubic is a heuristic algorithm that is driven by events at the sender side, including ACK receipt, packet loss, etc. The policy of Cubic is based on packet loss events, which leads to poor performance in scenarios where packet loss is present. PBQ is a combination of the DRL method PPO and BBR with the advantages of both. Our deployment adopts the client/server (C/S) mode, which only modifies QUIC in client and adopts the standard QUIC implementation in the server. We deeply integrated PBQ into the QUIC client. We added the Monitoring module for network state statistics, the Decision module for rate control, and the Pacer module for rate distribution. The Decision and Pacer modules are asynchronously executed, and they do not affect the behavior of the clients.

We specifically designed states and actions. The Monitoring module periodically collects the state of the environment and sends the processed data to the Decision module. We collected a number of environment states, including pacing rate, current RTT and minimum RTT, interval, packet loss rate, CWnd, and so on. The output of the Pacer module includes the CWnd and pacing rate of the senders. The BBR regulates the pacing rate of the clients, and the DRL agent gives the congestion window as the predicted BDP. Then, the Pacer module changes the sending behavior of the clients.

## 4. Simulation Performance

We tested the PBQ-enhanced QUIC in various scenarios on the open-source network simulator ns-3 [23] and compared the results with those of QUIC using different congestion control algorithms. We wrote simulation scripts in C++ and implemented our DRL agent based on MindSpore in python. Then, we ran the comparison experiments on a Dell PowerEdge R840 server with 256 G memory, 64 cores, and a GeForce RTX 3090 GPU. We tested the model on a large number of links with different parameters and analyzed the sensitivity of PBQ-enhanced QUIC to the number of link flows and packet loss rate. When the testing and training links were quite different, the PBQ-enhanced QUIC still performed well.

### 4.1. Our Simulation Environment

The application of DRL methods is inseparable from the interaction with the environment. Several researchers have implemented their own simulation environments, but they all target the TCP domain, and there is no mature solution for the simulation environment of the QUIC protocol alone. At the same time, we think that the development of a network simulation environment is a huge and meticulous project that should be focused on the study of congestion control algorithms; therefore, we chose the ns-3 platform to build the training and testing environment. We built our own simulation environment based on NS3-gym [24] and Quic-NS-3 [25] to test PBQ-enhanced QUIC.

### 4.2. Training

We also applied the PPO method to the congestion control part of QUIC. We trained PPO and PBQ on the link shown in Figure 7. For training, we set N to two, which means that two traffic flows shared a link. The link parameters are shown in Table 2. Our model converged in 50 epochs, each consisting of 1200 steps. In contrast, the experimental results showed that after combining with BBR, the model converged faster, and the throughput and RTT performance were more stable in the initial interaction.

As illustrated in Figure 8b,c, the algorithm we propose can effectively improve the stability of the pretraining action while ensuring the quality of the link. As shown in Figure 8a, the combination with the BBR effectively speeds up the speed of convergence, which solves the problems we described before.

### 4.3. Testing

Similarly, we compared PBQ-enhanced QUIC with QUIC using current learning-based congestion control algorithm Remy and heuristic algorithms such as Bic [26], Cubic, Low Extra Delay Background Transport (LEDBAT) [27], NewReno [28], Vegas [29], and BBR. [id = S.L.] The link parameters are shown in Table 3.

In traditional algorithms, BBR is based on network modeling, while the remaining algorithms take packet loss events as congestion occurrence signals. In this part, we still used the dumbbell link in Figure 7. To demonstrate the adaptability of our algorithm, there was a large deviation between our test and training links. Specifically, we determined the performance of PBQ-enhanced QUIC in different scenarios, including its excellent adaptation and tolerance to packet loss and number of flows.

### 4.4. Packet Loss Rate

We tested the proposed algorithm using a previously trained model in a different network scenario. The throughput changes with the packet loss rate as illustrated in Figure 9a, and the RTT is shown in Figure 9b.

When packet loss occurs, QUIC with congestion control algorithms that take packet loss events as a signal, such as QUIC with Cubic and QUIC with NewReno, frequently enter fast recovery, resulting in low throughput and low RTT. While QUIC with BBR requires constant probing of the network, QUIC with Remy relies on a state-action table. It is difficult for them to balance throughput and RTT. Our algorithm can efficiently identify the overall state of the network and can output relevant actions by periodically collecting the network state. As the packet loss rate increases, the throughput performance of our algorithm is less affected, and it is the best among the different packet loss rates. Furthermore, our proposed algorithm achieves higher throughput analso has better packet loss tolerance ability than QUIC with Remy and QUIC with BBR.

### 4.5. Flow Number

Then, the packet loss rate was set to 2.5%, and we modified the number of flows and compared the throughput and delay performance of different QUIC implementations. The results are shown as Figure 10a,b. Figure 10a represents the total link throughput. Figure 10b indicates the average RTT.

Owing to the presence of random packet loss, algorithms that take packet loss events as congestion occurrence signals fail to accurately identify the cause of congestion. In the policies of these congestion control methods, the current link can only support a low throughput; thus, the throughput is at a low level. In this case, the flows barely affect each other. As a result, as shown in Figure 10a, the total throughput of the link linearly increases with the number of flows. They cannot accurately estimate the link BDP and occupy the full link bandwidth, so the average link RTT is close to the default RTT. Compared with QUIC with BBR and QUIC with Remy, PBQ-enhanced QUIC reduces the detection process of link BDP and achieves excellent throughput and RTT by optimizing the reward function.

## 5. Conclusions

In this study, we first developed a two-level regulatory mechanism PBQ, which combines the heuristic algorithms BBR and DRL to approximate PPO. The convergence rate of the model is accelerated by using BBR to estimate the specific transmission rate. Moreover, we proposed PBQ-enhanced QUIC, an implementation of QUIC that uses PBQ as a congestion control algorithm. Unlike QUIC with heuristic congestion control algorithms, our QUIC implementation learns congestion control rules from experience by using RL signals. Therefore, our QUIC implementation can be better adapted to various network settings.

As shown in Section 4.2, the combination with BBR can effectively speed up the convergence of the PPO during the training phase on the premise of ensuring link quality. As shown in Section 4.2 and Section 4.4, compared with other QUIC versions, PBQ-enhanced QUIC achieves higher throughput performance in various network settings. PBQ-enhanced also has better RTT performance than QUIC with BBR and QUIC with Remy. We think that combining DRL methods with traditional algorithms to design congestion control mechanisms for QUIC will be a major trend in the future, and PBQ-enhanced QUIC provides a new idea to do so.

In future work, we plan to build a testbed in real-world networks and test PBQ-enhanced QUIC on it. We will improve the PBQ-enhanced QUIC based on its performance in real-world networks. Moreover, different applications have different requirements on network metrics, so we plan to design congestion control algorithms for different applications by taking advantage of the feature that the congestion control part of QUIC is implemented in user mode. We expect different applications to perform efficiently on the same network. 

## Figures and Tables

**Figure 1 entropy-25-00294-f001:**
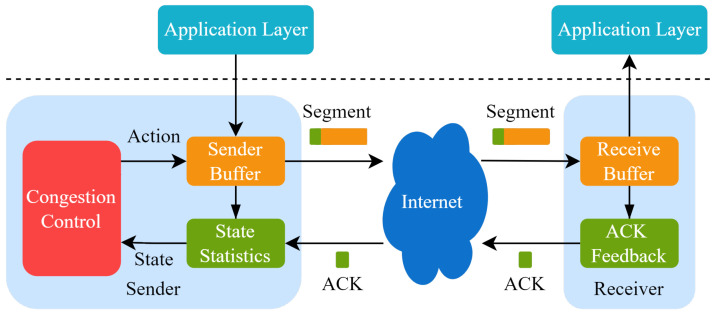
A schematic of congestion control in TCP.

**Figure 2 entropy-25-00294-f002:**
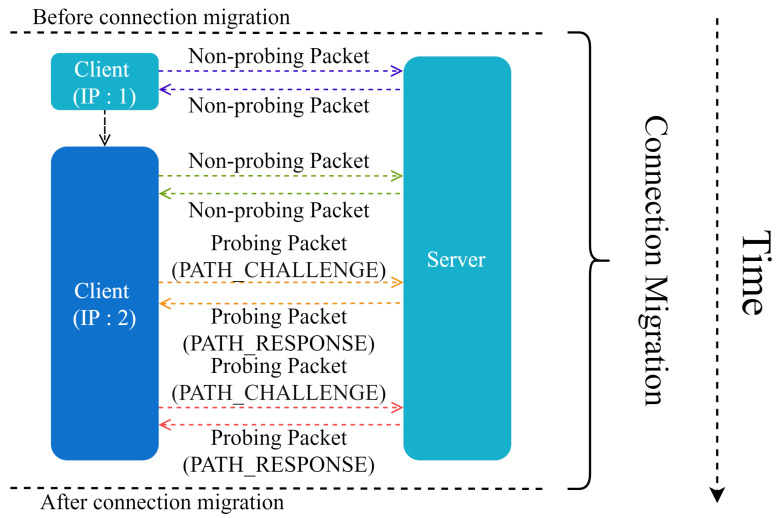
Connection migration.

**Figure 3 entropy-25-00294-f003:**
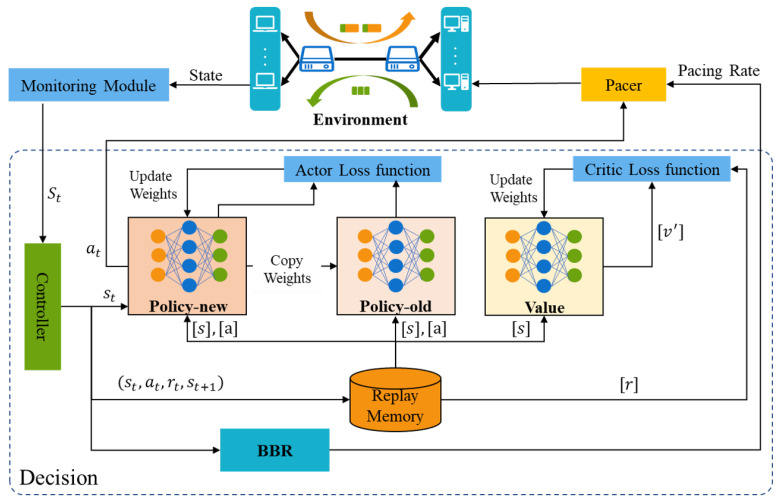
The framework of PBQ.

**Figure 4 entropy-25-00294-f004:**
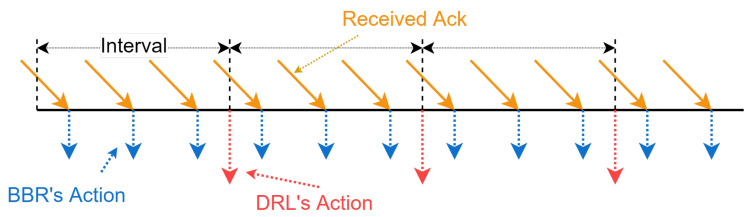
The two-level control logic of PBQ’s enhanced QUIC.

**Figure 5 entropy-25-00294-f005:**
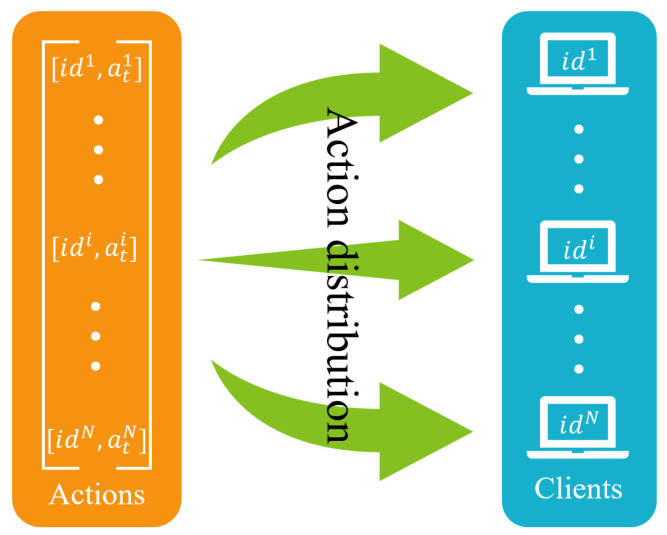
Action distribution in the Pacer module.

**Figure 6 entropy-25-00294-f006:**
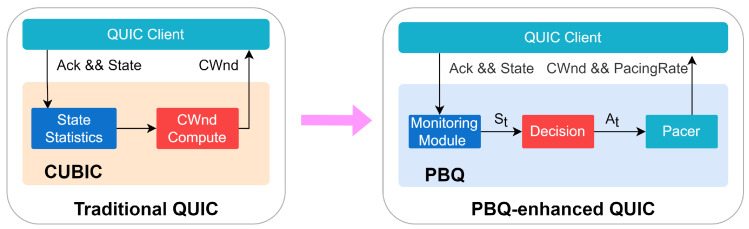
PBQ-enhanced QUIC.

**Figure 7 entropy-25-00294-f007:**
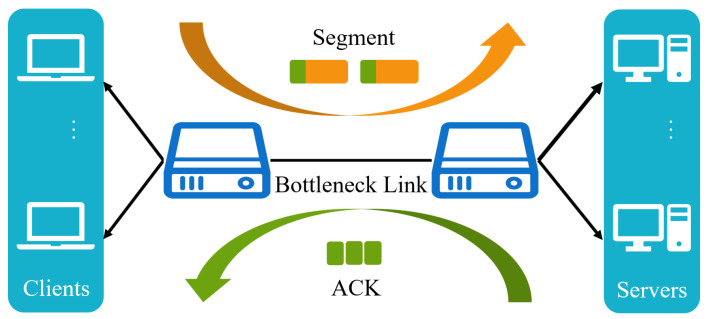
The simulation link of PBQ.

**Figure 8 entropy-25-00294-f008:**
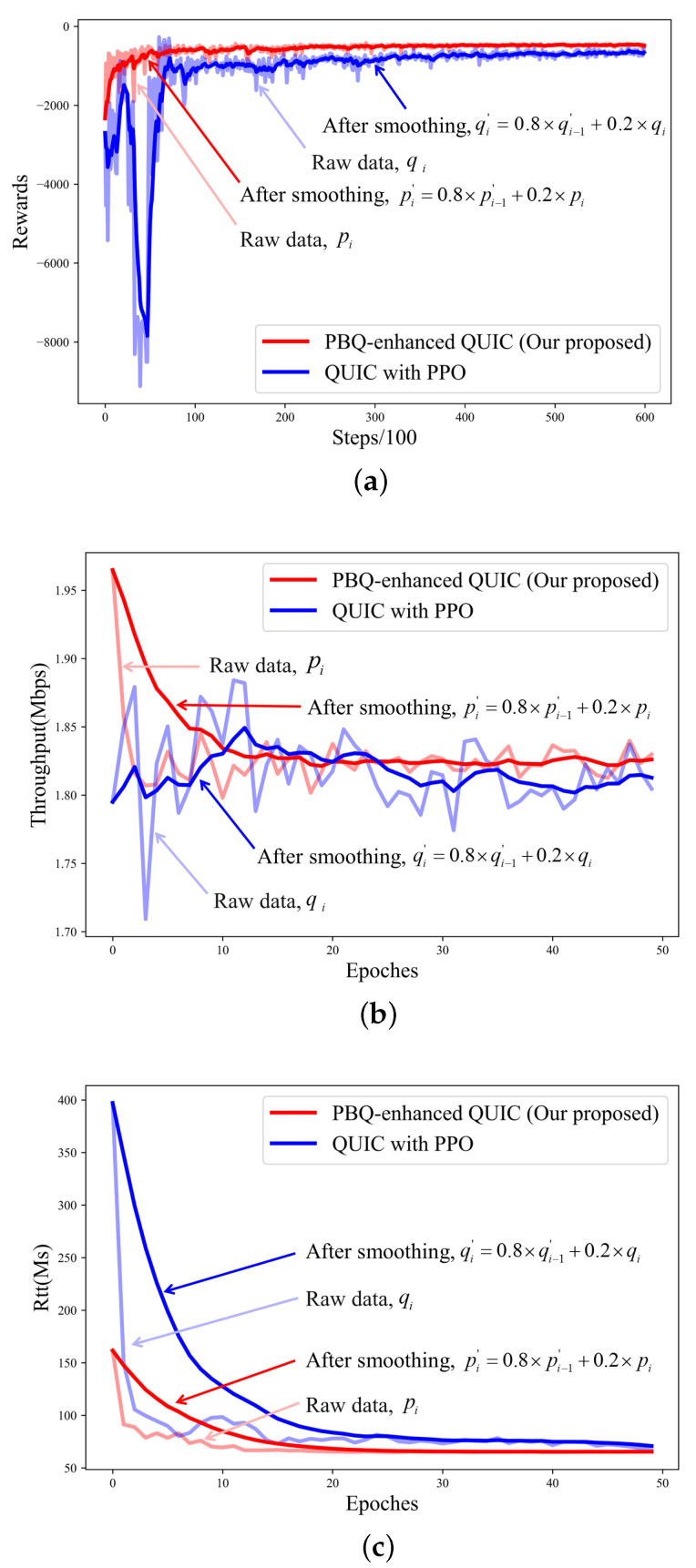
Training process after 600 episodes: (**a**) training reward; (**b**) training throughput; (**c**) training RTT.

**Figure 9 entropy-25-00294-f009:**
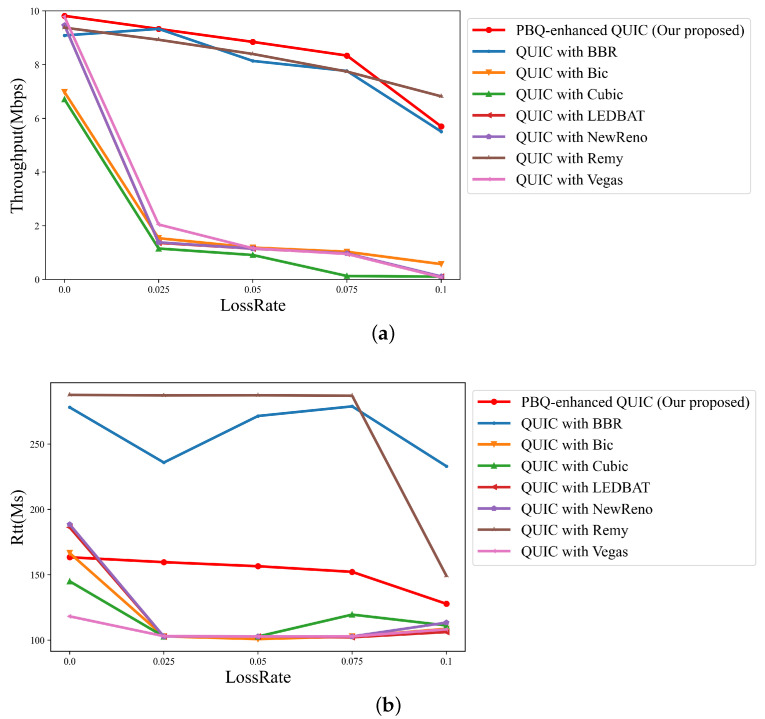
Link quality of schemes across different packet loss rates with flow number 2: (**a**) throughput; (**b**) RTT.

**Figure 10 entropy-25-00294-f010:**
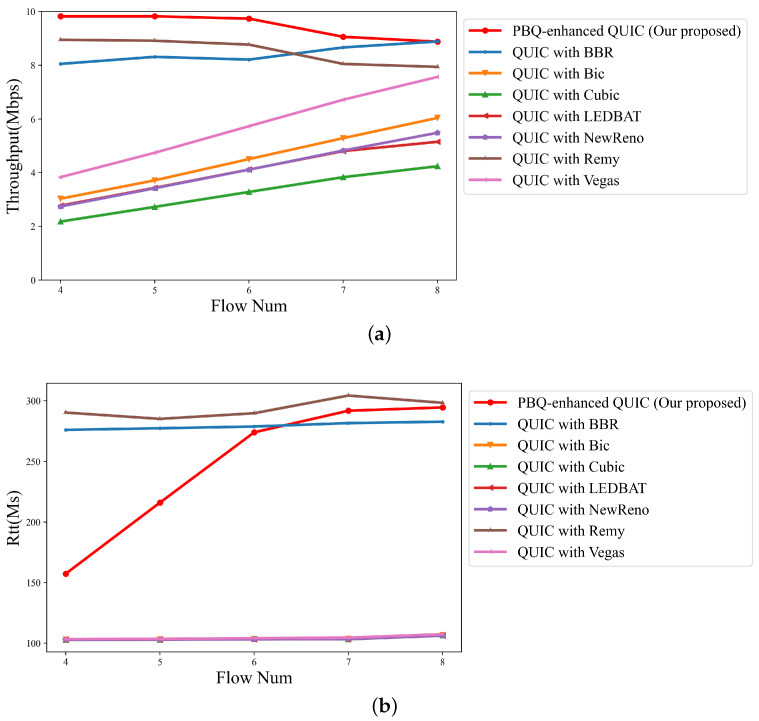
Link quality of schemes for different flow numbers with packet loss rate of 2.5%: (**a**) throughput; (**b**) RTT.

**Table 1 entropy-25-00294-t001:** State statistics description.

State	Description
$CWnd_{t}$	Current congestion window
$interval_{t}$	Cata update interval
$deliveryRate_{t}$	Average delivering rate (throughput)
$RTT_{t}$	Averaged RTT
$packetLoss_{t}$	Average loss rate of packets

**Table 2 entropy-25-00294-t002:** Training link.

Attribute	Value
Number of flows	2
Bottleneck bandwidth	2 Mbps
RTT	30 ms
Queue capacity	75 Kilobytes
Queue scheduling algorithm	First Input First Output (FIFO)

**Table 3 entropy-25-00294-t003:** Testing link.

Attribute	Value
Number of flows	2∼8
Bottleneck bandwidth	10 Mbps
RTT	100 ms
Packet loss rate	0%∼10%
Queue capacity	75 Kilobytes
Queue scheduling algorithm	FIFO

## Data Availability

No new data were created or analyzed in this study. Data sharing is not applicable to this article.

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
