# Peer review of "PBQ-Enhanced QUIC: QUIC with Deep Reinforcement Learning Congestion Control Mechanism"

_entropy, 2023, doi:10.3390/e25020294_

Round 1
Reviewer 1 Report
In this paper, the authors proposed an efficient congestion control mechanism by using the deep reinforcement learning. The proposed algorithm combines the traditional Bottleneck Bandwidth and Round-trip propagation time (BBR) with Proximal Policy Optimization (PPO). Since the PPO agent can recognize the network state and estimates the link, the algorithm was evaluated by metrics like Bandwidth-Delay Product (BDP) as congestion window (CWnd), and the BBR specifies the sending rate of the client. Experiment results show that the proposed PBQ-enhanced QUIC achieves much better performance in both throughput and RTT than existing popular versions of QUIC, such as QUIC with Cubic and QUIC with BBR.
The proposed algorithm is novel and efficient. The presentation is satisfying. Thus, the reviewer suggests the publication of the paper.
a minor suggestion is thatthe title of subsection 4.4 and 4.5 maybe revised by removing the ‘friendliness’.
Reviewer 2 Report
In general, it is strongly recommended that the presented article be better organized. In addition, some of my comments are below.
1) In the introduction, state clearly what the problem is and what is the main reason for doing this study.
2) Focus on the existing studies with their pros and cons.
3) Abstract section not well prepared.
4) Let the flow diagram of the proposed method be drawn.
5) Has the channel status and management mechanism in the network been overlooked?
6) How are alpha and beta values determined? How the tuning of similar features was done. You can use the below reference as an example. https://doi.org/10.1016/j.compbiolchem.2021.107619
7) learning algorithm should be explained better and in detail.
8) What kind of result should we expect if another algorithm (RR, SPN, etc.) is used instead of the queue scheduling algorithm? What is your comment?
9) Rewrite the conclusion section. Also, include information about future works.
Reviewer 3 Report
The authors combine DRL methods with traditional algorithms to design congestion control mechanism for QUIC, named "PBQ-enhanced QUIC" providing potential enhancements in both RTT and throughput, which is a promising idea. The authors also mention the limitations of the study.
Author Response
Dear Editor and Reviewer:
Thanks for your comments concerning our manuscript entitled “PBQ-enhanced QUIC: QUIC with A Deep Reinforcement Learning Congestion Control Mechanism” (ID: entropy-2134128). Your comments are very helpful for improving our paper. We have carefully checked the spelling and grammar, and have improved the English writing in the revised manuscript. The modified portions are marked in blue. We hope that this manuscript will meet with approval. Again, many thanks for your comments.
Round 2
Reviewer 2 Report
The requested corrections and comments were answered at the expected level.